# Potential Mechanisms of Metformin-Induced Apoptosis in HeLa Cells

**DOI:** 10.3390/biom13060950

**Published:** 2023-06-06

**Authors:** Zhaoli Chu, Yao Tan, Chenxing Xu, Dongting Zhangsun, Xiaopeng Zhu

**Affiliations:** 1Key Laboratory of Tropical Biological Resources of Ministry of Education, School of Pharmaceutical Sciences, Hainan University, Haikou 570228, China; 20100700210003@hainanu.edu.cn; 2Medical School, Guangxi University, Nanning 530004, China; 2228402007@st.gxu.edu.cn (Y.T.); 2228391051@st.gxu.edu.cn (C.X.)

**Keywords:** metformin, HeLa cells, RNA-seq, pharmacological mechanism

## Abstract

Metformin is a traditional antidiabetic drug that also shows potential antitumor effects in cervical cancer. However, some of its apoptosis-related mechanisms are still unclear. In this study, flow cytometry, western blotting, and RNA sequencing (RNA-seq) were used to evaluate the molecular mechanisms of metformin in HeLa cells. The results showed that metformin inhibited cell viability and promoted apoptosis, the protein expression level of Caspase-3 (CASP3) was increased and that of BCL-2 was decreased in HeLa cells treated with metformin. The RNA-seq results indicated a total of 239 differentially expressed genes between the metformin and control check (CK) groups, with 136 genes upregulated and 103 genes downregulated, and 14 of them were found to be associated with apoptosis signaling pathways. The *DDIT3* and *HRK* genes were robustly upregulated in HeLa cells by the endoplasmic reticulum (ER) stress and the mitochondrial pathway of apoptosis. Metformin also affects the expression of *PPP2R5C*, *PPP2R5A*, and *RRAGA*, which participate in biological processes such as PI3K-AKT, mTOR, and AMPK signaling pathways. Metformin mediates the expression of related genes to induce apoptosis.

## 1. Introduction

Metformin, a member of the traditional class of drugs for type 2 diabetes, is currently one of the most frequently used glucose-lowering medications in clinical practice because of its stable tolerance and safety [1]. Metformin lowers blood sugar levels, which may be related to a reduction in hepatic glucose production [2]. Recently, many studies have shown that metformin also has other clinically beneficial effects. Chen J. et al. indicated that metformin can increase the *C. elegans* lifespan through the lysosomal pathway [3] and slow the ovarian aging process, probably by reducing oxidative damage [4], and low-dose metformin has been found to alleviate human cellular aging [5]. Metformin enhances autophagy and restores mitochondrial function [6]. In addition, many studies have shown that metformin inhibits cell proliferation in several cancers, such as breast cancer, liver cancer, colon cancer, pancreatic cancer, skin cancer, prostate cancer, ovarian cancer, lung cancer, and cervical cancer [7]. Scheen A. J. et al. revealed that the mechanism by which metformin reduces tumor incidence is AMP-activated protein kinase (AMPK)-dependent or AMPK-independent and suppresses the production of nuclear factor kappa B (NF-κB). Metformin promotes apoptosis by inhibiting NF-κB and increasing the expression of activated transcription factor-3 (ATF-3) [8,9]. Among the various potential antitumor mechanisms of metformin identified, AMPK-dependent mechanisms are of pivotal importance. These underlying molecular mechanisms result in the inhibition of protein synthesis and unfolded protein production, cell cycle arrest, activation of the immune system, and damage to cancer stem cells [10,11]. Many studies have revealed that metformin can be used to treat cancer. Table 1 shows the pharmacological mechanisms of metformin in various tumors.

Cervical cancer is one of the most prevalent cancers worldwide. It is estimated that 604,000 women were affected by cervical cancer worldwide in 2020, according to the International Agency for Research on Cancer (IARC). Previous studies have shown that metformin activates the AMPK/p53 axis and suppresses PI3K/AKT signaling in cervical cancer cells, which leads to the induction of apoptosis. Moreover, metformin induces cell cycle arrest, apoptosis, and autophagy. Metformin inhibits proliferation and promotes apoptosis. Although metformin has shown promising therapeutic potential against cervical tumors, its antitumor mechanism is complex and elucidating the molecular pharmacological mechanism of cervical tumor cells is urgently needed.

## 2. Materials and Methods

### 2.1. Cell Lines and Materials

The human cervical cancer cell line HeLa was purchased from the Kunming Institute of Zoology (Yunnan, China) and the cell line number is KCB86019YJ. The cell line name is human cervical cancer cell line HeLa, the species is human, and the tissue origin is the cervical cancer gland. Metformin was purchased from SigmaAldrich (St Louis, MO, USA). The Cell Counting Kit-8 was purchased from Sangon Biotech Co., Ltd. (Shanghai, China). The Annexin V-FITC/PI apoptosis kit was purchased from MultiSciences (Lianke) Biotech Co., Ltd. (Hangzhou, China).

### 2.2. Cell Culture and Viability Assay

HeLa cells were cultured in Dulbecco’s Modified Eagle Medium (DMEM, Sangon Biotech, Shanghai, China) supplemented with 10% fetal bovine serum (FBS, Procell Life Science and Technology, Wuhan, China) in a humidified incubator at 37 °C with 5% CO_2_. When the HeLa cells were 80–90% confluent, they were dissociated in 0.25% trypsin–EDTA solution (Sangon Biotech, Shanghai, China) and then plated at the same density each time. HeLa cells were seeded in a 96-well plate (2 × 10^3^ cells/well) in DMEM. HeLa cells were treated with metformin (0, 3, 6, 12, 24, and 48 mM) for 24 h. Following cell treatment, 10 µL of CCK−8 solution was applied to each well and incubated at 37 °C for 1.5 h. The absorbance was measured at a wavelength of 450 nm using a Spectra Max M2 Microplate Reader (Molecular Devices, CA, USA). Cell viability was quantified using the formula (cell viability (%) = [(OD_450_ of the experimental group) − OD_450_ of the blank group)/(OD_450_ of the CK group − OD_450_ of the blank group)] × 100%). All experimental data were obtained from at least three independent measurements.

### 2.3. Apoptosis Analysis

HeLa cells were seeded in a 6-well plate (3 × 10^5^ cells/well) in DMEM. HeLa cells were treated with metformin (0, 3, 12, and 24 mM) for 24 h. HeLa cells were fixed with precooled 4% paraformaldehyde for 15 min at room temperature, and 10 µL (5 µg/mL) of DAPI (Beyotime, Shanghai, China) solution was applied to each well and incubated for 1 h. Then the cells were immediately analyzed with a Cytation 1 Cell Imaging Multimode Reader (BioTek, VT, USA).

HeLa cells (3 × 10^5^ cells/well) were seeded in 6-well plates in DMEM and treated with metformin (0, 3, 12, and 24 mM) for 24 h. The cells were collected after treatment with 0.25% trypsin (without EDTA), washed twice with PBS, and added to 500 µL of 1× binding buffer; the cell count was adjusted (1 × 10^4^ cells/group), and the cells were stained with 5 µL of Annexin V-FITC and 10 µL of PI (MultiSciences, Hangzhou, China) in the dark at room temperature for 5 min prior to flow cytometry analysis (Millipore, Darmstadt, IN, USA). All experimental data were obtained from at least three independent measurements.

### 2.4. Western Blotting

HeLa cells were lysed in 80 µL RIPA buffer (Beyotime, Shanghai, China) containing 0.8 µL 1 mM fluoride, and the protein concentration of each sample was determined using the BCA Protein Assay Kit (Bioss, Beijing, China). Proteins (20 µg) were boiled at 95 °C for 5 min, followed by separation by 15% SDS–PAGE. After electrophoresis, the proteins were electrophoretically transferred to PVDF membranes at 100 mA for 90 min. The membranes were blocked with 5% nonfat milk for 1 h at room temperature and incubated overnight at 4 °C with the following primary antibodies: CASP3, BCL−2, beta Actin (β−actin) (Proteintech Group, Inc., Chicago, IL, USA). The catalog numbers are 66470-2-Ig,60178-1-Ig, and 66009-1-Ig. Next, the membranes were treated for 1 h at room temperature with horseradish peroxidase (HRP)-conjugated secondary antibody (Proteintech Group, Inc., Chicago, IL, USA). The catalog number is SA00001-1. The membranes were visualized with enhanced chemiluminescence solution (Biosharp, Beijing, China) (A liquid:B liquid = 1:1) using an Amersham ImageQuant 800 system (Cytiva, Tokyo, Japan).

### 2.5. RNA-Seq to Identify Apoptosis-Related Signaling Pathways

HeLa cells (5 × 10^5^ cells/mL) were cultured in 6-well plates and treated with metformin (0 and 24 mM) for 24 h. Total RNA was isolated from HeLa cells using a FastPure Cell/Tissue Total RNA Isolation Kit (Vazyme Biotech Co., Ltd., Nanjing, China) according to the manufacturer’s instructions, and the integrity and concentration were determined and evaluated by a Nanodrop 2000 spectrophotometer (Thermo Scientific, Waltham, MA, USA) to ensure the quality of sequencing samples. Total RNA was sequenced by Biomarker Technologies Co. (Beijing, China). The cDNA library was built by Biomarker Technologies, and it was sequenced utilizing the Illumina platform. Gene sequencing data that had been pruned and cleaned was compared with human reference genomes (Homo_sapiens.GRCh38_release95.genome.fa) using HISAT2 software. Differentially expressed genes (DEGs) were analyzed with the R software packages edgeR 3.32.1 and DESeq2 1.30.1. The difference in the expression of a DEG between the treatment groups was expressed as the log2 (fold change) value. Fold change ≥ 2 and FDR < 0.01 were utilized as the DEG screening criteria. The Spearman correlation coefficient (R) was the metric used to evaluate recurrent correlations in biology. The DEG and gene clustering data were shown on the ordinate, while the sample name and clustering results were shown on the abscissa. The Kyoto Encyclopedia of Genes and Genomes (KEGG) pathway database was used as the basis to analyze significant enrichment, and the top 20 pathways with the lowest significant Q value using the hypergeometric test are displayed.

### 2.6. Identification of Key Genes by RT-qPCR

The RNA-Seq data were verified by RT–qPCR analysis of key genes, including *CASP3*, *BCL-2*, *HRK*, and *DDIT3*. A HiScript II Q RT SuperMix for qPCR kit (Vazyme Biotech Co., Ltd., Nanjing, China) was used to reverse transcribe 1 µg of total RNA into cDNA following the manufacturer’s instructions. The 2 × ChamQ Universal SYBR qPCR Master Mix Kit (Vazyme Biotech Co., Ltd., Nanjing, China) was used for RT-qPCR. *ACTB* was used as the internal reference gene, and the expression levels of the key genes in each sample were normalized to that of the reference gene by the ∆C_T_ method to analyze the significance of differences. RT-qPCR primers Appendix A were synthesized by Sangon Biotech Co., Ltd. (Shanghai, China).

### 2.7. Statistical Analysis

All statistical analyses were performed using GraphPad Prism Software 8.0 (GraphPad Software, San Diego, California, USA). FlowJo™ software (Becton, Dickinson and Company, California, USA) was used for flow cytometry analysis. ImageJ 1.8.0.172 (National Institutes of Health, Bethesda, MD, USA) was used for the densitometric analysis of western blots. Protein band exposure and contrast were adjusted simultaneously for all groups and the parameters were consistent. The data were presented as the mean ± S.D values and were analyzed by one-way ANOVA. Dunnett’s *t*-test was used to compare the two groups, and ns indicates that the difference was not statistically significant, while *p* < 0.05 was considered statistically significant. Each experiment was repeated at least three times.

## 3. Results

### 3.1. Metformin Inhibited HeLa Cell Viability

The effects of different concentrations (0, 3, 6, 12, 24, and 48 mM) of metformin on HeLa cells were determined. The results showed that metformin reduced cell viability in a concentration- and time-dependent manner (Figure 1). The cell viability rate was greater than 70% when the metformin concentration was less than 12 mM. The cell viability rate was less than 50% when the metformin concentration was increased to 48 mM at 24 h and 48 h. When HeLa cells were treated with metformin for 12 h, 24 h, and 36 h, the IC_50_ values of metformin were 47.34 ± 9.94 mM, 35.87 ± 2.11 mM, and 35.13 ± 4.20 mM, respectively Appendix A. According to the above experimental results, the metformin treatment time was identified to be 24 h, and the concentrations of metformin used for the subsequent treatment of HeLa cells were 3, 12, and 24 mM (low, medium, and high concentrations, respectively).

### 3.2. Metformin-Induced Apoptosis in HeLa Cells

To determine whether metformin affects the apoptosis of HeLa cells, HeLa cells were stained with DAPI solution and Annexin V-FITC/PI or PI/RNase. The number of dark-blue nuclei was increased in the metformin group compared to the CK group (Figure 2A). Flow cytometry results revealed that 99.50% of the cells were located in Q4 (Annexin V−/PI−) in the CK group (0 mM) and that these cells were normal viable cells. The number of HeLa cells in the early apoptotic quadrant Q3 (Annexin V+/PI−) and late apoptotic quadrant Q2 (Annexin V+/PI+) was significantly increased in a dose-dependent manner (Figure 2B). The apoptosis rate increased considerably, from 0.46 ± 0.10% (0 mM) to 91.30 ± 3.48% (3 mM), 92.07 ± 3.43% (12 mM), and 95.03 ± 2.13% (24 mM) at metformin concentrations of 3, 12, and 24 mM, respectively (Figure 2C).

### 3.3. Metformin Influenced the Levels of CASP3 and BCL-2 in HeLa Cells

To determine whether metformin affects CASP3 and BCL-2, we measured the protein levels of CASP3 and BCL-2 (Figure 3A). Compared with the control treatment, 24 mM metformin increased CASP3 expression but decreased BCL-2 expression in HeLa cells. In particular, the expression of CASP3 in the metformin group was upregulated by 63.35 ± 32.15% (*n* = 3) (Figure 3B), and the expression of BCL-2 in the metformin group was downregulated by 74.93 ± 20.22% (*n* = 3) (Figure 3C).

### 3.4. Metformin Impacted the Expression of Genes in HeLa Cells, as Determined by RNA-Seq

RNA-seq was performed on four biologically reproducible control and treated (24 mM metformin) samples to explore the mechanism of metformin-induced apoptosis in HeLa cells. A total of 50.17 Gb clean reads and approximately 5.82 Gb of data per sample were generated by RNA-seq analysis, and more than 92.8% of the bases had a quality score of Q30 Appendix A in the sample correlation analysis. Samples of the same treatment were typically clustered together, and there were noteworthy differences between various treatment groups (Figure 4A). A total of 239 significant DEGs were identified in metformin-treated HeLa cells (Figure 4B). Among them, 136 were upregulated and 103 were downregulated, and 14 genes were determined to be associated with apoptosis (Figure 4C, Appendix A. Additionally, we applied KEGG pathway annotation and enrichment analysis to explore the pathways to which these 14 genes were linked (Figure 4D). The analysis results showed that the top 10 pathways ranked by *p*-value were as follows: biosynthesis of amino acids; neurotrophin signaling pathway; apoptosis; selenocompound metabolism; arginine biosynthesis; glycine, serine and threonine metabolism; cholesterol metabolism; cysteine and methionine metabolism; arginine and proline metabolism; and glutathione metabolism. The majority of these pathways influence apoptosis indirectly by altering the metabolism and synthesis of amino acids, cholesterol, selenium compounds, and glutathione. Notably, the *HRK* and *DDIT3* genes were directly related to the apoptotic pathway; *HRK*-encoded proteins interact with BCL-2 and BCL-X(L), and *DDIT3* mediates endoplasmic reticulum (ER) stress and leads to apoptosis. In summary, metformin mediates the apoptosis of HeLa cells by affecting mitochondria, ER-related apoptotic pathways, and nutrient metabolism-related genes.

### 3.5. Verification of Gene Expression

The RNA-seq results showed that the *HRK* and *DDIT3* genes were upregulated upon treatment with 24 mM metformin (Figure 5A). The KEGG pathway analysis was performed to map the relevant signaling pathways, and RT-qPCR was used to pinpoint the critical genes in the pathway. *HRK*, *DDIT3*, and *CASP3* were upregulated by 137.50 ± 34.48% (*n* = 4), 793.40 ± 162.60% (*n* = 4), and 137.20 ± 35.48% (*n* = 4), respectively. However, the *BCL-2* gene showed no discernible difference (Figure 5B). The *HRK* and *DDIT3* genes indirectly or directly affect the expression of BCL-2/BCL-XL, BAX/BAK complexes on the mitochondrial membrane, and indirectly affect the expression of CASP family genes, such as *CASP3*, *CASP6*, *CASP9* genes, and eventually leading to apoptosis (Figure 5C). Metformin also affected the expression of *PPP2R5C*, *PPP2R5A*, and *RRAGA* (Figure 5D). The fragments per kilobase million (FPKM) of *PPP2R5C* gene in the experimental group was 41.29 ± 0.70 (*n* = 4), and the FPKM of the *PPP2R5A* gene was 22.55 ± 0.47 (*n* = 4), which were adjusted upward compared with the FPKM value of the CK group. The FPKM of *RRAGA* was 77.32 ± 3.59 (*n* = 4), which was statistically significant compared with that of the CK group.

## 4. Discussion

Metformin is a biguanide drug that is considered the first-line therapy for type 2 diabetes mellitus. Based on recent studies, metformin inhibits the proliferation of cancer cells and elucidates related mechanisms in different types of cancer, including lung cancer, ovarian cancer, prostate cancer, breast cancer, and cervical cancer. The potential molecular mechanisms of metformin in cancer treatment have been extensively studied. Metformin’s antitumor effect is multifaceted: metformin has been found to significantly reduce the in vitro growth and invasion capacities of A549 and H1651 cells [21], inhibit ROS-TFE3-dependent autophagy [17], decrease the production of Mesothelin (MSLN), downregulate IL-6/STAT3 signaling activity, and induce apoptosis in ovarian cancer cells [22]. In addition, to prevent prostate cancer cells from migrating and invading other tissues, metformin may induce the upregulation of PEDF in prostate cancer cells [14]; metformin also inhibits the mTOR signaling pathway, which mediates the inhibitory effect of dipeptidyl peptidase-4 (DPP-4) inhibitors on breast cancer. However, the anticancer mechanism of metformin in cervical cancer is still unclear.

Cervical cancer is the fourth most common malignancy in women and one of the most prevalent gynecologic health problems. Thus, there is a need to further elucidate the mechanisms of effective therapies for cervical cancer. The report by Xia C et al. in 2020 suggested that metformin enhanced apoptosis and prevented migration through the AMPK/p53 and PI3K/AKT pathways in human cervical cancer cells [23]. The study showed that metformin has potential therapeutic effects on HeLa cells [24]. The present study indicated that 12 mM and 24 mM metformin can inhibit HeLa cervical cancer cell viability. Consistent with the results of the present study, Hakimee H. et al. reported that metformin significantly reduced cell proliferation, and after treating HeLa cells for 48 h with 20 mM metformin, the cell viability rate was less than 20%. In contrast, the present study indicated that metformin demonstrated cytotoxicity against HeLa cells, as determined by the measured IC_50_ value of 35.87 ± 2.11 mM (*n* = 3). After exposure to 24 mM metformin for 24 h, cell viability was noticeably reduced. We measured cell viability using the CCK-8 assay, which reflects the number of viable cells in metformin-acting HeLa cells. However, it is unclear whether metformin is toxic to HeLa cells, or the result of cell viability, or induces apoptosis. It suggests that metformin may induce apoptosis in HeLa cells, which we continue to investigate. Apoptosis is a form of programmed cell death and plays an important role in cancer therapy as it is one of the important strategies to combat malignant tumors. The mechanism of apoptosis is multifaceted and involves many signaling pathways and proteins. The CASP family and BCL-2 family play an important role in the regulation of apoptosis. Zhao H. et al. demonstrated that, whereas metformin therapy decreased the expression of BCL-2 in pancreatic cancer cells, it increased the expression of CASP3 and BAX [25]. Metformin can promote AGS cell apoptosis via BCL-2 downregulation, an effect that is related to the AMPK signaling pathway and mTOR/AKT [26]. The present study showed that, compared with that in the CK group, CASP3 expression was significantly increased by 58.45 ± 19.90% (*n* = 3) and BCL-2 expression was significantly decreased by 75.79 ± 24.28% (*n* = 3) in HeLa cells treated with metformin. The three replicates were consistent Appendix A. The results suggested that metformin affected crucial pathways by affecting proteins from the BCL-2 family and the CASP family, which enhanced apoptosis.

Transcriptome analysis has been used to find a potential mechanism of metformin action on cervical cancer from the perspective of gene functions and pathways. To determine the potential pathways underlying the mechanisms of metformin in HeLa cells, we completed an RNA-seq analysis. As we were specifically interested in apoptosis, genes associated with apoptosis were further investigated to identify the related apoptotic pathways. Findings from the RNA-seq study indicated that metformin treatment of HeLa cells resulted in the dysregulation of mitochondrial and ER stress-mediated pathways related to apoptosis. Furthermore, metformin treatment changed the expression of apoptosis-related genes (including *NKX2-5*, *BEX2*, *CHAC1*, *ARG2*, *DDIT3*, *EDN2*, *FOSL1*, *PHLDA1*, *CTH*, *ANGPTL4*, *GDF15*, *G0S2*, *TP73*, and *HRK*), especially *HRK* and *DDIT3*. Previous studies have reported that *HRK* is a key player in GBM cell death and regulates apoptosis by modulating the levels of the antiapoptotic proteins BCL-2 and BCL-X(L) [9]. In a similar vein, *DDIT3* plays a critical role in inducing ER stress-related apoptosis and overexpressing *DDIT3* results in cell cycle arrest and apoptosis in osteoclasts [27]. In addition, *DDIT3* promotes cancer stem cell stemness by upregulating CEBPβ in gastric cancer [28]. However, the effect of metformin on *HRK* and *DDIT3* expression has not been reported in HeLa cells. Our results demonstrated that metformin promoted *HRK* and *DDIT3* expression in HeLa cells. RNA-seq analysis of metformin-treated HeLa cells revealed potential alterations of *HRK* and *DDIT3*. We hypothesized that metformin influences the expression of *CASP3* and *BCL-2* in HeLa cells, hence promoting *DDIT3*-induced ER stress. The interaction of *DDIT3* with the apoptosis inhibitors BCL-2 and BCL-X(L) via its BH3 domain to affects apoptosis. The KEGG pathways related to *HRK* and *DDIT3* were examined, their apoptotic routes were plotted, and RT–qPCR was used to confirm the expression levels of key genes in the pathways. We found that the mRNA expression levels of *HRK*, *DDIT3*, and *CASP3* were consistent with the RNA-seq results. The novelty of our study is that we identified two genes associated with apoptosis; *HRK* and *DDIT3*. Some studies revealed that *HRK*, as a potential pro-apoptotic gene, may contribute to the development and progression of many human cancers, such as prostate cancers and astrocytic tumors [29]. In addition, it was previously reported that *HRK* is expressed in normal tissue but is decreased in melanoma. Artificial overexpression of *HRK* by recombinant adenovirus-induced caspase-dependent apoptosis inhibited melanoma cell growth in vitro. The *HRK* gene encodes an important proapoptotic mitochondrial protein of the BCL-2 family. However, the effect of metformin on *HRK* gene expression in cervical cancer cells has not been fully elucidated [30]. Our study fills a gap in the *HRK* gene in HeLa cervical cancer cells and provides a theoretical basis for the regulation of mitosis in mitochondria by BCL2 family proteins [31]. In addition, the *DDIT3* gene is related to the endoplasmic reticulum stress signaling pathway. The study found that through the microarray screening, *DDIT3* showed abnormally expressed in gastric cancer tissue compared to normal tissues. In the same way, the effect of metformin on *DDIT3* gene expression in cervical cancer cells has not been fully elucidated. Metformin affected the expression of *PPP2R5C* [32], *PPP2R5A* [33], and *RRAGA* genes in HeLa cells, and found that these genes were associated with signaling pathways such as AMPK, mTOR, and PI3K/AKT Appendix A. In addition, some tumor-related genes, such as *BEX2*, *EDN2*, *ANGPTL4*, and *TP73* genes [34,35], were also involved in tumor suppression and transcription, tumorigenesis, tumor cell invasion, and encoding p53 family members. The *CTH* [36] gene encodes an enzyme that converts cystathionine from methionine to cysteine, while the CASP protein family is aspartate proteolytic enzymes containing cysteine, an enzyme that may be involved in the synthesis of CASP protein, thereby regulating apoptosis.

In summary, our study revealed that metformin can enhance the apoptosis of HeLa cells and that metformin functions by upregulating *HRK* and *DDIT3* expression. These results indicated that metformin may be used as a potential agent to promote ER stress and the interaction of *DDIT3* with the apoptosis inhibitors BCL-2 and BCL-X(L) via the BH3 domain. *PPP2R5C*, *PPP2R5A*, and *RRAGA* genes were associated with signaling pathways such as AMPK, mTOR, and PI3K/AKT. This study provides a theoretical framework for the prospective use of metformin to treat cervical cancer in the future.

## Figures and Tables

**Figure 1 biomolecules-13-00950-f001:**
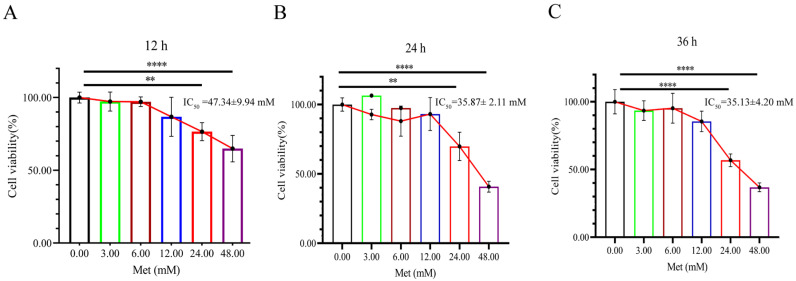
Metformin inhibited HeLa cell viability. HeLa cells were treated with 0 (CK), 3, 6, 12, 24, and 48 mM metformin for 12 h (**A**), 24 h (**B**), and 36 h (**C**), respectively, and the Cell Counting Kit-8 assay was utilized to determine the cell viability of HeLa cells. Met represents metformin. The data were represented as the mean ± S.D. The multivariate comparative analysis Dunnett’s *t*-test was used for statistical analysis, *n* = 5, ** *p* < 0.01, **** *p* < 0.0001 vs. CK group.

**Figure 2 biomolecules-13-00950-f002:**
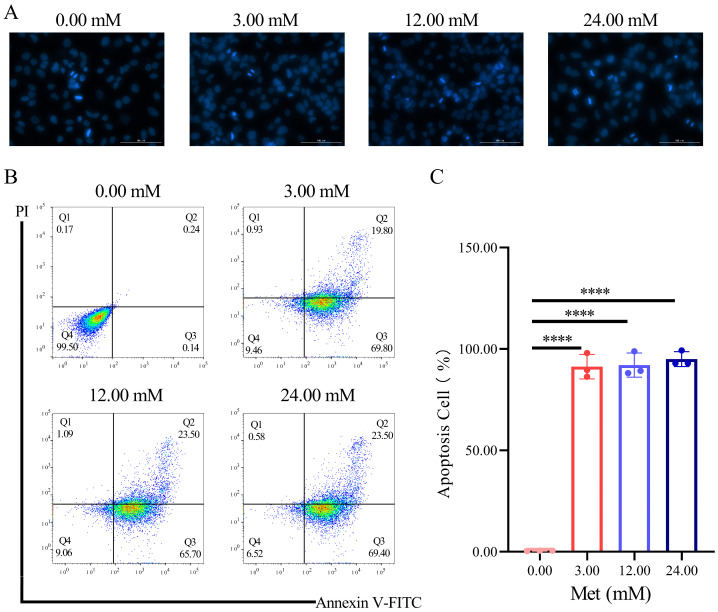
Metformin-induced apoptosis in HeLa cells. HeLa cells were treated with metformin (0, 3, 12, and 24 mM) for 24 h. Cell apoptosis was detected by DAPI staining (scale: 100 µm) (**A**) and flow cytometry (**B**). Analysis using flow cytometry of three biological duplicate sets (**C**). Met represents metformin and metformin concentrations were represented by various colors. The data were represented as the mean ± S.D. The 0 mM metformin group is the CK group. The multivariate comparative analysis Dunnett’s *t*-test was used for statistical analysis, *n* = 3, **** *p* < 0.0001 vs. CK group.

**Figure 3 biomolecules-13-00950-f003:**
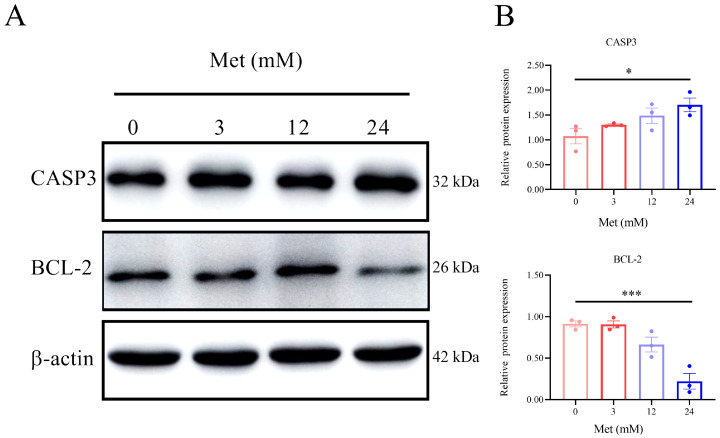
Metformin influenced levels of CASP3 and BCL-2 in HeLa cells. HeLa cells were treated with metformin (0, 3, 12, and 24 mM) for 24 h. The protein expressions of CASP3 and BCL-2 were analyzed by Western blot (**A**) and semi-quantified in CASP3 (**B**) and BCL-2 (**C**) proteins. Met represents metformin, and metformin concentrations were represented by different colors. The data were represented as the mean ± S.D. The 0 mM metformin group is the CK group. The multivariate comparative analysis Dunnett’s *t*-test was used for statistical analysis, *n* = 3, * *p* < 0.05, *** *p* < 0.001 vs. CK group.

**Figure 4 biomolecules-13-00950-f004:**
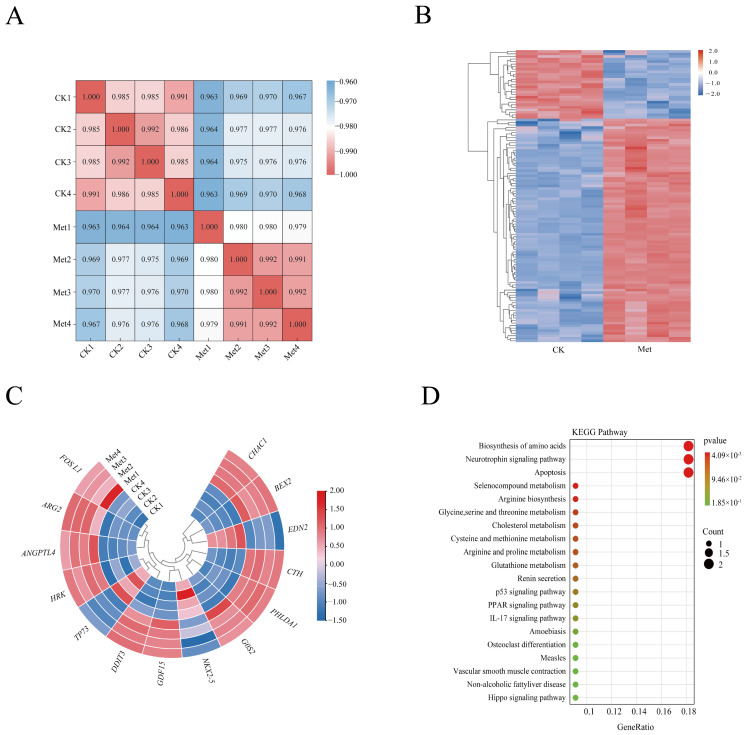
Metformin impacted the expression of genes in HeLa cells by RNA-seq. Correlation analysis of all samples; red denotes a strong correlation, while blue denotes a low correlation (**A**); DEGs heat maps for control check (CK) and metformin groups. Grouping of genes and samples based on expression similarity (**B**). Heatmap of expression of overlapped 14 genes and Scatter plots of KEGG pathways for 14 distinct genes (**C**,**D**). Met represents metformin. The size of the dot represents the number of genes, while the color represents the range of *p*-values.

**Figure 5 biomolecules-13-00950-f005:**
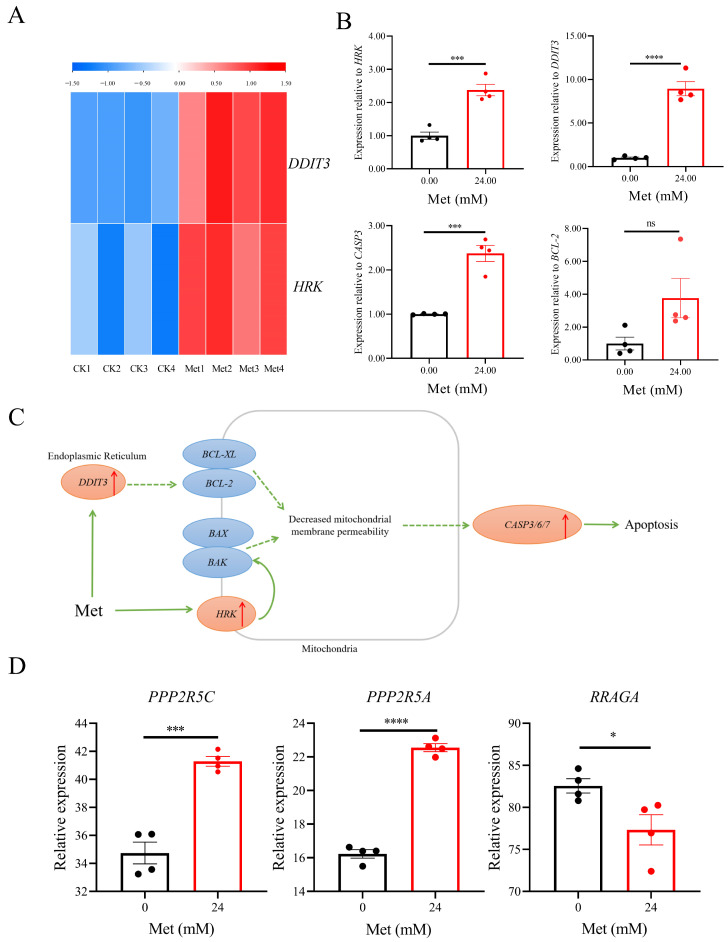
Verification of gene expression. Heatmap of expression of overlapped *HKR* and *DDIT3* genes (**A**); the expression level of *HKR* and *DDIT3* genes was identified by using RT-qPCR (**B**); the signal pathways of *HRK* and *DDIT3* genes on the KEGG pathway were analyzed, and the pathway map was drawn by KEGG (**C**); the expression level of *PPP2R5C*, *PPP2R5A*, and *RRAGA* genes was identified by using RNA-seq (**D**). Met represents metformin and metformin concentrations were represented by different colors. The data were represented as the mean ± S.D. The 0 mM metformin group is the CK group. The multivariate comparative analysis Dunnett’s *t*-test was used for statistical analysis, *n* = 4, * *p* < 0.05, *** *p* < 0.001, **** *p* < 0.0001 vs. CK group, ns indicates no statistically significant difference.

**Table 1 biomolecules-13-00950-t001:** The mechanism of metformin with different tumor cells.

Tumor Types	Targets or Pathway	References
Liver cancer	mTOR, AMPK pathways	[12]
Pancreatic cancer	miRNA, cancer stem cells	[13]
Prostate cancer	Pigment Epithelium-derived Factor (PEDF),	[14]
Breast cancer	mTOR pathway ROS-TFE3-dependent autophagy	[15,16,17]
Cervical cancer	Focal adhesion kinase (FAK), protein kinase B (PKB), Ras-related C3 botulinum toxin substrate (RAC1) protein	[17]
	AMPK O-GlcNAcylation	[17]
	AMPK/p53 and PI3K/AKT pathways	[18]
	Cyclin D1 and P53 expression	[19]
	Liver kinase B1 (LKB1)	[20]

## Data Availability

The datasets presented in this study can be found in online repositories. the SRA records will be accessible with the following link after the indicated release date: https://www.ncbi.nlm.nih.gov/sra/PRJNA964600 (accessed on 1 June 2023).

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
