# Peer review of "Potential Mechanisms of Metformin-Induced Apoptosis in HeLa Cells"

_biomolecules, 2023, doi:10.3390/biom13060950_

Round 1

Reviewer 1 Report

Metformin, a first-line drug for diabetes mellitus type 2, appears to be associated with a lower risk and improved outcomes in cervical cancer suggesting that metformin may have the potential to be used as an adjuvant in cancer therapy. Zhaoli Chu et al. investigated the molecular mechanisms of metformin in HeLa cells and concluded that this drug inhibits cell proliferation and promotes apoptosis by increasing the protein expression level of Caspase-3 (CASP3) and decreasing BCL-2. The study is well designed and adequately described. Since this is an interesting outcome from the use of metformin for cancer therapy, I would suggest to go deeply in details and continue to investigate the effect of this drug on cancer cells by proposing other studies on it.

As a comment I would like to ask the authors why they used Cell Counting Kit-8 assay to determine cell viability instead of the MTT. Furthermore, they measured cell viability but they discussed about cell proliferation. If they really want to check cell proliferation I would like to suggest to use BrdU assay that is more appropriate in this case.

English language is fine. Minor editing of English language are required.

Reviewer 2 Report

The manuscript by Chu et al., entitled: Potential mechanisms of metformin-induced apoptosis in HeLa cells describe the mechanistic details of Metformin in HeLa cells. 

Overall, the study is well-designed. However, I have a few concerns. Metformin is very well characterized for its mode of action including its role in apoptosis and autophagy. So I failed to understand the novelty of this study. Authors should elaborate more on what this study adds to existing literature and the missing gaps filled. 

The other repeated western blot images are of better quality. Authors should consider replacing the figures. 

Which software was used for the densitometric analysis of western blots?

The method should contain the antibody catalog numbers for each antibody used. 

Protein level validation/confirmation of a few RNASeq targets will be helpful.

Round 2

Reviewer 2 Report

The authors have made considerable changes and the quality of the revised manuscript is improved